# CCR7 Mediates Dendritic-Cell-Derived Exosome Migration and Improves Cardiac Function after Myocardial Infarction

**DOI:** 10.3390/pharmaceutics15020461

**Published:** 2023-01-30

**Authors:** Youming Zhang, Wei Gao, Jie Yuan, Xin Zhong, Kang Yao, Rong Luo, Haibo Liu

**Affiliations:** 1Department of Cardiology, Qingpu Branch of Zhongshan Hospital, Fudan University, Shanghai 201700, China; 2Department of Cardiology, Shanghai East Hospital, School of Medicine, Tongji University, Shanghai 200100, China; 3Shanghai Institute of Cardiovascular Diseases, Zhongshan Hospital, Fudan University, Shanghai 200030, China

**Keywords:** CCR7, dendritic cells, exosomes, CD4^+^ T cells, myocardial infarction

## Abstract

Dendritic cells (DCs) play key roles in promoting wound healing after myocardial infarction (MI). Our previous studies have shown that exosomes derived from DCs (DEXs) could migrate to lymphoid tissue and improve cardiac function post-MI by activating CD4^+^ T cells; however, the mechanism of DEXs’ migration to lymphoid tissue and the improvement of cardiac function are still unknown. In our study, we found that CCR7 expression significantly increased in MI-DEXs compared with control-DEXs; meanwhile, CCL19 and CCL21, the ligands of CCR7, significantly increased in the serum of MI-model mice. Subsequently, we overexpressed and knocked down CCR7 in MI-DEXs and found that overexpressed CCR7 enhanced the migration of MI-DEXs to the spleen; however, CCR7 knockdown attenuated MI-DEXs’ migration according to near-IR fluorescence imaging. Furthermore, overexpressed CCR7 in MI-DEXs enhanced the MI-DEXs’ improvement of cardiac function after MI; however, CCR7-knockdown MI-DEXs attenuated this improvement. In addition, after DEXs’ migration to the spleen, MI-DEXs activated CD4^+^ T cells and induced the expression of IL-4 and IL-10, which were significantly increased in the MI-DEX group compared with the control group. In conclusion, CCR7 could mediate DEXs’ migration to the spleen and improve cardiac function after MI, and we found that the mechanism was partly via activation of CD4^+^ T cells and secretion of IL-4 and IL-10. Our study presented an innovative method for improving cardiac function by enhancing the migration ability of MI-DEXs after MI, while CCR7 could be a potential candidate for MI-DEX bioengineering to enhance migration.

## 1. Introduction

Though considerable progress has been made in the diagnosis and treatment of myocardial infarction (MI), which is still the leading cause of death worldwide [1], the progression of MI often leads to adverse ventricular remodeling, eventually resulting in ventricular dysfunction and death [2,3,4]. The immune system plays a crucial role in the ventricular remodeling process and contributes to both the inflammatory and reparative phases [5,6,7]. Our previous study showed that exosomes derived from dendritic cells (DCs) could migrate to lymphoid tissue and improve cardiac function post-MI by activating CD4^+^ T cells [8]. Prolonging the retention time of DC-derived exosomes (DEXs) could result in better consequences for immunoregulation and enhance its therapeutic effects [9]. However, whether MI-DEXs’ migration could affect their therapeutic effects in addition to the mechanism of DEXs’ migration to lymphoid tissue is still unclear.

As the central immunoregulators in innate and adaptive immunity, DCs play crucial roles in facilitating wound healing after MI [10]. Mice depleted of DCs induced severe inflammatory reactions and worse cardiac remodeling after MI compared with the control group, which suggested that DCs have played protective roles in the pro-inflammatory response to MI [11]. In addition, infarct lysate-primed DCs activated infarct tissue-specific regulatory T cells in lymph nodes, spleen, and infarcted myocardium and facilitated wound remodeling, preserved left ventricular systolic function, and improved survival after MI [12,13]. These results suggest DCs play a protective role after MI and that their therapeutic effect is partly because of CD4^+^ T cell activation. 

Exosomes are small vesicles secreted by cells that contain proteins, DNA, and RNA and are considered messengers for cell-to-cell information exchange [14]. Exosomes from immune cells engage in intercellular communication in immune cells [15], while DEXs could also participate in antigen presentation [16] and migrate to lymphoid tissues after MI [8,9]. 

Chemokine receptors (CCR) and their ligands (CCL) are key factors in immune cells’ migration. DCs highly express CCR7 after capturing antigens [17]. CCR7, together with its ligands, CCL19 and CCL21, promote the accurate migration of DCs to lymphocyte tissues [18]. However, whether CCR7 contributes to DEXs’ migration is still unknown. 

In our study, we explored the role of CCR7 and its ligands CCL19/CCL21 in DEXs’ migration, including the crosstalk between DEXs’ migration and its therapeutic effects on cardiac function after MI, and provided a new candidate for protecting cardiac function after MI.

## 2. Materials and Methods

### 2.1. Animals

We purchased wild-type, male C57BL/6 mice from the Shanghai Laboratory Animal Center, which we used for bone marrow-derived dendritic cell (BMDC) culture and making MI models. We raised all the mice without pathogens with a 12:12-h light–dark cycle, 22 ± 1 °C room temperature, 65–70% humidity, and free access to food and water. The Institutional Review Board of Zhongshan Hospital of Fudan University and the Shanghai Institutes of Biological Sciences-CAS (A5894-01, January 2017) approved this study, and the animal research complied with the standards of animal welfare in China.

### 2.2. MI Model Induction and Treatments

We used a rodent ventilator (Harvard Bioscience, Inc., Cambridge, MA, USA) for mouse ventilation and an isoflurane delivery system, and we used 2% isoflurane inhalation (RWD Life Science, Shenzhen, China) (Harvard Bioscience, Inc., Cambridge, MA, USA) for anesthetizing. Then, we opened the mouse chests and exposed the hearts. We ligated the left coronary artery (LCA) with an 8–0 silk suture. We observed the blanching of the myocardium in the infarcted area, which we determined to be myocardial infarction. We used mice receiving the same operation but without LCA ligation as the sham operation group. Finally, we evacuated the air, sutured the skin, and put the mice back in the cage before waiting for them to wake up. After 24 h of MI, we injected different DEXs or saline via the mouse tail vein.

### 2.3. Cell Isolation and Culture

We isolated BMDCs from the femurs of C57BL/6 mice following methods described in our previous study (8). Briefly, we sacrificed the mice and isolated the femurs; then, we washed bone marrow cells from the femurs and cultured them at 10^6^ cells/mL in BD Falcon cell culture flasks (BD Biosciences, New York, NY, USA) in RPMI-1640 (Gibico, New York, NY, USA) containing 10% fetal bovine serum (FBS, Life Technologies, New York, NY, USA), 1% Penicillin/Streptomycin (Gibico, New York, NY, USA), 1 ng/mL IL-4, and 10 ng/mL granulocyte–macrophage colony-stimulating factor (both from R&D Systems, Minneapolis, MN, USA). We gently washed off non-adherent cells after 48 h and cultured the remaining loosely adherent clusters with media changes every 2 days. On day 7, we used medium without exosomes, and we collected and determined BMDCs using anti-mouse CD11c, CD40, CD80, and CD86 antibodies via flow cytometry. We collected the BMDCs’ supernatants for DEX isolation.

We isolated splenic CD4^+^ T cells from the spleens of different groups of mice using the EasySep^TM^ Mouse CD4^+^ T Cell Isolation Kit and following the manufacturer’s instructions. 

### 2.4. Lentivirus Production and Infection

We purchased mouse CCR7 lentivirus, mouse CCR7 shRNA lentivirus, vector lentivirus, and scramble shRNA lentivirus from Genomeditech (Shanghai, China) Co., Ltd. We infected BMDCs with various plasmids using lentivirus according to the manufacturer’s instructions.

### 2.5. Mimic MI Microenvironment

There were three methods to mimic the MI microenvironment, which were described in detail in our previous study [8]: (I) the supernatants of hypoxic primary rat cardiomyocytes; (II) the supernatants of infarcted mouse myocardium; (III) the supernatants of necrotic HL-1 cells. All three methods activate DCs in mice to a similar extent as MI. In the present study, we used the supernatants of necrotic HL-1 cells to mimic the MI microenvironment. The methods are described in detail in our previous study [8].

### 2.6. DEX Isolation and Characterization

We used ExoQuick-Tc™ Exosome Precipitation Solution (System Biosciences Inc., Palo Alto, CA, USA) for the isolation of DEXs according to the manufacturer’s instructions. We used CD63 and Alix to identify the DEX via Western blotting. We determined particle morphology via H7500 transmission electron microscopy (Hitachi, Tokyo, Japan). We determined particle size distributions via the Nanosight NS300 (Malvern, Malvern, UK).

### 2.7. Experimental Grouping

DEX groups: We divided DEX into four groups: (1) DEX group: DEX isolated from BMDCs treated with supernatant from normal cardiomyocytes; (2) MI-DEX group: DEX isolated from BMDCs treated with 100 μL/mL of supernatant from necrotic cardiomyocytes for 24 h; (3) CCR7^over^-MI-DEX group: DEX from BMDCs treated with mouse CCR7 lentivirus for 72 h and then 100 μL/mL of supernatant from necrotic cardiomyocytes for 24 h; and (4) CCR7^down^-MI-DEX group: DEX from BMDCs treated with mouse CCR7 shRNA lentivirus for 72 h and then 100 μL/mL of supernatant from necrotic cardiomyocytes for 24 h.

Animal groups: We divided the animals into six groups: (1) sham group: mice received the same surgical procedure but not ligation of the LCA; (2) MI group: mice received a saline injection after 24 h of MI; (3) DEX group: mice received a DEX injection after 24 h of MI; (4) MI-DEX group: mice received a MI-DEX injection after 24 h of MI; (5) CCR7^over^-MI-DEX group: mice received a CCR7^over^- MI-DEX injection after 24 h of MI; and (6) CCR7^down^-MI-DEX group: mice received a CCR7^down^-MI-DEX injection after 24 h of MI.

### 2.8. DEX Labeling and Fluorescence Imaging

We used DiR (Thermo Fisher Scientific, Waltham, MA, MA, USA) for labeling DEX. Then, we used exosome spin columns (MW 3000; Thermo Fisher Scientific, Waltham, MA, USA) to remove the dye and purify DEX. Finally, we used the IVIS Spectrum System (PerkinElmer, Houston, TX, USA) to acquire the near-IR fluorescence images and compare the radiant efficiency of the body, spleen, and liver among different groups.

### 2.9. Echocardiography

On day 14, we examined the cardiac function by echocardiography (Vevo 2100, VisualSonics, Toronto, ON, Canada). We used M-mode tracings to measure LV end-systolic diameter (LVIDS) and LV end-diastolic diameter (LVIDD). We calculated percent fractional shortening (FS%) and ejection fraction (EF%) using Vevo 2100 workstation software.

### 2.10. Histology

On day 14, we sacrificed mice and fixed hearts with 4% paraformaldehyde for 24 h. Then, we cut the paraffin-embedded sample into 5 µm thick tissue sections, dewaxed in xylene (Sangon Biotech co., Ltd., Shanghai, China), rehydrated using ethanol, stained the sections using Masson’s trichrome reagent, and finally sealed the sections using neutral gum. We calculated the wall thickness of the infarct zone and the fibrosis percentage of sections using Image J V1.53e software under an optical microscope. For each section, we took three measurements of different parts of the infarct area for thickness analysis.

### 2.11. Flow Cytometry Analysis

We used an Fc receptor blocker (CD16/32, BD Bioscience, New York, NY, USA) to incubate the sample to reduce non-specific antibody binding. We resuspended the freshly isolated samples and then stained them using CD45-AlexaFlour700 (Biolegend, San Diego, CA, USA), CD3-FITC (Biolegend, San Diego, CA, USA), and CD4-APC (Biolegend, San Diego, CA, USA) for 30 min at 4 °C. We performed flow cytometry using the CytoFLEX Platform (BeckmanCoulter, Brea, CA, USA) and analyzed the data with the CytExpert V2.4.0.28 software (BeckmanCoulter, Brea, CA, USA).

### 2.12. Quantitative Real-Time Polymerase Chain Reaction

We used a FastPure Cell/Tissue Total RNA Isolation Kit V2 (Vazyme Biotech, Nanjing, China) to extract total RNA from cells or tissue according to the manufacturer’s instructions. We synthesized cDNA by reverse transcription from 500 ng of total RNA using the HiScript III RT SuperMix cDNA Synthesis Kit (Vazyme Biotech, Nanjing, China) according to the manufacturer’s instructions. Then, we diluted the cDNA to 10:1 and amplified using the ChamQ Universal SYBR qPCR Master Mix Kit (Vazyme Biotech, Nanjing, China) following the manufacturer’s instructions. We incubated the reaction in a 96-well plate at 95 °C for 3 min, followed by performing 40 PCR cycles at 95 °C for 5 s and 60 °C for 34 s on an Applied Biosystems 7500 Real-time PCR System (Thermo Fisher Scientific, Inc., Waltham, MA, USA). We collected and analyzed data using the 2^ΔΔCt^ method. First, we normalized the values of genes against GAPDH; then, we compared with the experimental controls. We synthesized primers at Tiangen Biochemical Technology Co., Ltd., and the sequences are listed below. TNF-α-F: AGAAACACAAGATGCTGGGACAGT; TNF-α-R: CCTTTGCAGAACTCAGGAATGG; IFN-γ-F: ATGAACGCTACACACTGCATC; IFN-γ-R: CCATCCTTTTGCCAGTTCCTC; IL-1β-F: GCAACTGTTCCTGAACTCAACT; IL-1β-R: ATCTTTTGGGGTCCGTCAACT; IL-6-F: TAGTCCTTCCTACCCCAATTTCC; IL-6-R: TTGGTCCTTAGCCACTCCTTC; IL-4-F: GGTCTCAACCCCCAGCTAGT; IL-4-R: GCCGATGATCTCTCTCAAGTGAT; IL-10-F: ACTGGCATGAGGATCAGCAG; IL-10-R: CTCCTTGATTTCTGGGGCCAT.

### 2.13. Statistical Analysis

Results are shown as mean ± SD (standard deviation). We used unpaired t-tests for the statistical analysis of the differences between two groups. We used one-way ANOVA followed by post hoc Tukey’s multiple tests for the statistical analysis of the differences among three or more groups. We performed all the statistical analyses using GraphPad Prism V8.0.1. We considered *p* < 0.05 as statistically significant.

## 3. Results

### 3.1. CCR7 Was Highly Expressed in MI-DEXs as well as Its Ligand CCL19/21 in MI-Mouse Spleen

Previously, studies showed that CCR7 and its ligands, CCL19 and CCL21, promoted the accurate migration of DCs to lymphocyte tissues [18]. To investigate whether CCR7 also contributed to DEXs’ migration, we generated MI-DEXs for further exploration. Figure 1A shows the ultrastructure of MI-DEXs via transmission electron microscopy (TEM). Figure 1B shows the expression of exosomal markers (CD63 and Alix). Figure 1C shows the particle size distributions of MI-DEXs determined by nanoparticle tracking analysis (NTA), and the mode diameter was 117 nm. Then, we explored CCR7 expression via Western blot. Our results showed that CCR7 was highly expressed in MI-DEXs (Figure 1D–E). Then, we also investigated its ligands, CCL19 and CCL21, in MI mouse serum and spleen (Figure 1F–J). We found that both CCL19 and CCL21 increased in MI mice, which indicated that MI-DEXs followed the same migration mechanisms as dendritic cells.

### 3.2. CCR7-Mediated MI-DEX Migration to the Spleen after MI

We found CCR7 was highly expressed in MI-DEXs. To further identify CCR7′s role in MI-DEXs’ migration, we first compared the migration of DEXs or MI-DEXs via near-IR fluorescence imaging after MI. Figure 2A,B show that both DEXs and MI-DEXs migrated to the spleen and liver after MI; however, more MI-DEXs migrated to the spleen compared with DEXs on day 1 after injection. However, there was no significant difference between DEXs and MI-DEXs regarding migration to the liver. This indicated that CCR7 may affect the migration of MI-DEXs, mediating their movement to the spleen after MI. To further identify CCR7′s role in MI-DEXs’ migration, we knocked down or overexpressed CCR7 in BMDC individually, and our results showed that the overexpressed CCR7 MI-DEXs could increase MI-DEX migration to the spleen after MI, while CCR7 knockdown significantly impaired this progress. Our results indicated that CCR7 mediates MI-DEXs’ migration, and CCR7 overexpression in MI-DEXs could enhance their migration to the spleen after MI.

### 3.3. Enhancing the Migration Ability of MI-DEXs Can Improve Cardiac Function after MI

Here, we explore the relationship between MI-DEXs’ migration and cardiac function. Our previous study showed that MI-DEXs improve cardiac function post-MI by activating CD4^+^ T cells in the spleen [8]. In addition, we found that overexpressed CCR7 could enhance the accumulation of MI-DEXs in the spleen via facilitating their migration; therefore, we needed to determine whether the accumulation of MI-DEXs would affect cardiac function post-MI. To do so, we used echocardiography and Masson staining to examine the cardiac function and remodeling on day 28 post-MI. The results showed that CCR7-overexpressed MI-DEXs received a better therapeutic effect, while CCR7 knockdown would impair the therapeutic effect (Figure 3A–E). Additionally, Masson staining results showed that MI-DEXs could retain LV-wall thickness but had no significant effect on fibrosis percentage (Figure 3F–H). In addition, though there was no significant difference among each MI-DEX group, the CCR7-knockdown group showed inferior LV-wall thickness, which may be the reason for its impaired echocardiography results.

### 3.4. MI-DEXs’ Activation of CD4^+^ T Cells in the Spleen

A previous study showed that the activation of CD4^+^ T cells played a key role in improving myocardial wound healing after MI [19]. To investigate the correlation between MI-DEXs and CD4^+^ T cells, we explored the CD4^+^ T cell percentage in the spleen’s leukocytes via flow cytometry analysis. Our results showed that CD4^+^ T cells were slightly decreased in the spleen in the MI-model group compared with the sham group; however, they markedly increased in the MI-DEX injection group (Figure 4). These results indicated that MI-DEXs could activate CD4^+^ T cells after MI, improving cardiac function.

### 3.5. MI-DEXs Induce an Anti-Inflammatory Cytokine Increase in Splenic CD4^+^ T Cells

We further explored the expression of cytokines in splenic CD4^+^ T cells in different groups and found the expression of IL-4 and IL-10 significantly increased in the MI-DEX group compared with the MI group (Figure 5). These results showed that injection of MI-DEXs could induce increases in the anti-inflammatory cytokines IL-4 and IL-10 in splenic CD4+ T cells after MI; moreover, the role of IL-4 and IL-10 as protective cytokines during cardiac remodeling after MI has been widely reported [20,21]. Thus, IL-4 and IL-10 could be functional cytokines for CD4^+^ T cells to protect cardiac function after MI. However, our results showed the expression of pro-inflammatory cytokines IFN-γ, IL-1β, and IL-6 was also significantly higher in the MI-DEX group, indicating a complex correlation between MI-DEXs and CD4^+^ T cells. Therefore, the underlying mechanism needs to be further explored. 

## 4. Discussion

Immune cells play a vital role in cardiac remodeling after MI, and our previous study showed that MI-DEXs could migrate to lymph tissues, such as the spleen and lymph nodes, improving cardiac function post-MI [8,9]. However, the mechanism of MI-DEXs’ migration to lymph tissues remains unknown. A previous study showed that CCR7 interacted with its CCL19 and CCL21 ligands and guided DC trafficking toward lymph nodes via afferent lymphatics [18]. DEXs, as the exomes secreted by DCs, may have followed the same migration pattern after MI. In this study, we found CCR7 was highly expressed in MI-DEXs, as well as its ligands, CCL19/21, in the MI-mouse spleen. We also explored the role of CCR7 in MI-DEXs’ migration to the spleen after MI and found that overexpressed CCR7 enhanced MI-DEXs’ migration to the spleen after MI, while down-regulated CCR7 expression reduced the accumulation of MI-DEXs in the spleen. Moreover, we found that MI-DEXs’ migration to the spleen could increase the infarct LV-wall thickness, decrease LV dilation, and improve cardiac function after MI. In addition, we explored the correlation between MI-DEXs and CD4^+^ T cells and found that MI-DEXs raised the percentage of CD4^+^ T cells in the spleen, as well as increasing the expression of anti-inflammatory cytokines IL-4 and IL-10 in splenic CD4^+^ T cells.

A previous study showed anti-inflammatory therapies could help with cardiac function recovery after MI [22], but the, inflammatory response is a double-edged sword in cardiac homeostasis and injury repair [10]. Therefore, it is important to find a proper way to mediate this process. Our previous study showed MI-DEXs could migrate to lymphoid tissue and improve cardiac function post-MI by activating CD4^+^ T cells [8]. Here, we focused on the mechanism of MI-DEXs’ migration and found that overexpressed CCR7 in MI-DEXs enhanced MI-DEXs’ migration to the spleen and improved cardiac function after MI, which indicated that modified MI-DEXs could be a useful candidate for post-MI non-cell therapy. Previous studies showed CCR7 could mediate inflammatory cells to participate in the inflammatory response [23], but our study demonstrated that CCR7 could mediate exosomes derived from DCs’ migration to the spleen after MI. 

We also explored the relationship between MI-DEXs and CD4^+^ T cells. Flow cytometry analysis and qPCR results showed that MI-DEXs increased the percentage of CD4+ T cells in the spleen and increased IL-4 and IL-10 expression in splenic CD4^+^ T cells. Previous research demonstrated the protective role of CD4^+^ T cells in scar formation and LV dilation after MI [19]. In addition, specific cardiac antigen-triggering healing CD4^+^ T cell activation was essential for cardiac-function recovery after MI [13]. Meanwhile, IL-4 and IL-10 were determined as protective cytokines for cardiac remodeling after MI [20,21]. Thus, our results indicated that MI-DEXs protected cardiac function via activation of CD4^+^ T cells secreting IL-4 and IL-10 after MI. Our findings provide a new way to protect cardiac function after MI.

Exosomes act as important mediators of cell-to-cell communication and play crucial roles in regulating immune cells and immune responses post-MI [15]. A previous study showed exosomes can facilitate the reparative process of infarcted myocardium, while the exosomes’ mechanism for preserving cardiac function was through the communication between lymphocytes and cardiac intrinsic cells [24]. Systemic delivery of immune-cell-derived exosomes is gradually being considered as a potent and effective new therapeutic option for healing damaged myocardium after MI [25]. In addition, because exosomes have natural material transportation properties and excellent biocompatibility characteristics, they have great potential as therapeutic drug delivery vesicles. Well-designed engineered exosomes could enhance their therapeutic effects, making them promising and inspiring tools for clinical application [26]. Our results demonstrated that CCR7 overexpression could enhance MI-DEXs’ migration to the spleen and improve cardiac function after MI. Thus, CCR7 could be a potential candidate for MI-DEX bioengineering to enhance its therapeutic effects after MI. 

However, there were some perplexing results in this study. We found increases in pro-inflammatory cytokine expression, such as IFN-γ, IL-1β, and IL-6, after MI-DEX injection in splenic CD4^+^ T cells. Due to the multiple subsets of CD4^+^ T cells existing in the spleen, including pro- and anti-inflammatory subsets, our results indicated that MI-DEXs increased anti-inflammatory cytokine expression in CD4^+^ T cells but did not affect pro-inflammatory cytokine expression. This phenomenon showed that the correlation between MI-DEXs and CD4^+^ T cells was complex and did not just convert CD4^+^ T cells into an anti-inflammatory type. The underlying mechanism needs to be further explored. In addition, we also did not find significant differences among different MI-DEX groups regarding LV fibrosis; however, large variations within each group may be the primary reason. 

## 5. Conclusions

In conclusion, DC-derived exosome migration to the spleen after MI was mediated by CCR7, and the mechanism of MI-DEX’s improvement of cardiac function included the activation of CD4^+^ T cells and secretion of IL-4 and IL-10. Our study presented an innovative way to improve cardiac function by enhancing the migration ability of MI-DEXs after MI, while CCR7 could be a potential candidate for MI-DEX bioengineering to enhance its migration and improve cardiac function after MI.

## Figures and Tables

**Figure 1 pharmaceutics-15-00461-f001:**
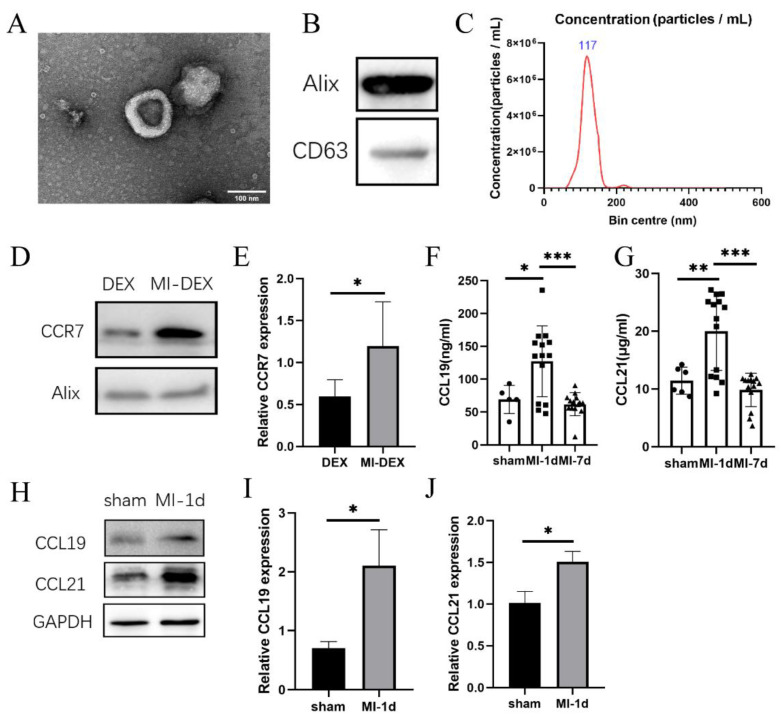
Characterization of MI-DEXs. Sample of MI-DEXs examined by transmission electron microscopy (TEM) image (**A**). CD63 and ALIX in MI-DEXs analyzed by Western blot analysis (**B**). The particle size distributions of MI-DEXs were analyzed by nanoparticle tracking, and the mode diameter was 117 nm (**C**). Expression of CCR7 in DEXs and MI-DEXs (**D**,**E**). *n* = 3–8. ELISA analysis of CCL19 and CCL21 in serum after MI (**F**,**G**). *n* = 5–15. Expression of CCL19 and CCL21 in the spleen after MI (**H**–**J**). *n* = 3–8, * *p* < 0.05, ** *p* < 0.01, *** *p* < 0.001.

**Figure 2 pharmaceutics-15-00461-f002:**
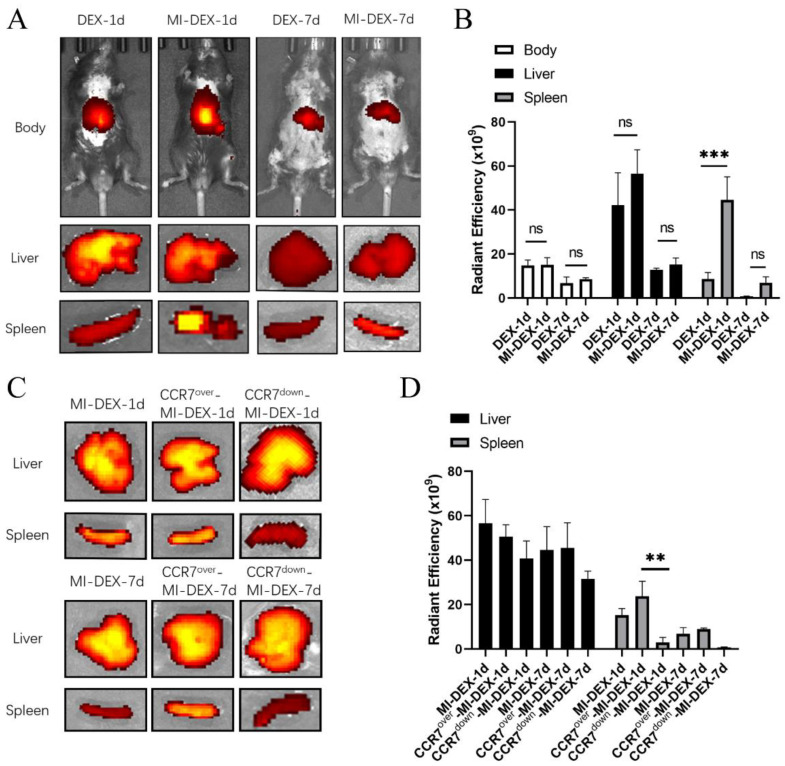
CCR7 mediates MI-DEXs’ migration. Representative near-IR fluorescence images of the main organs and body after DEXs or MI-DEX injections (**A**). Analysis of the radiant efficiency of body, spleen, and liver (**B**). Representative near-IR fluorescence images of the liver and spleen after different MI-DEXs injections (**C**). Analysis of the radiant efficiency of the liver and spleen after different MI-DEX injections (**D**). *n* = 3, ** *p* < 0.01, *** *p* < 0.001.

**Figure 3 pharmaceutics-15-00461-f003:**
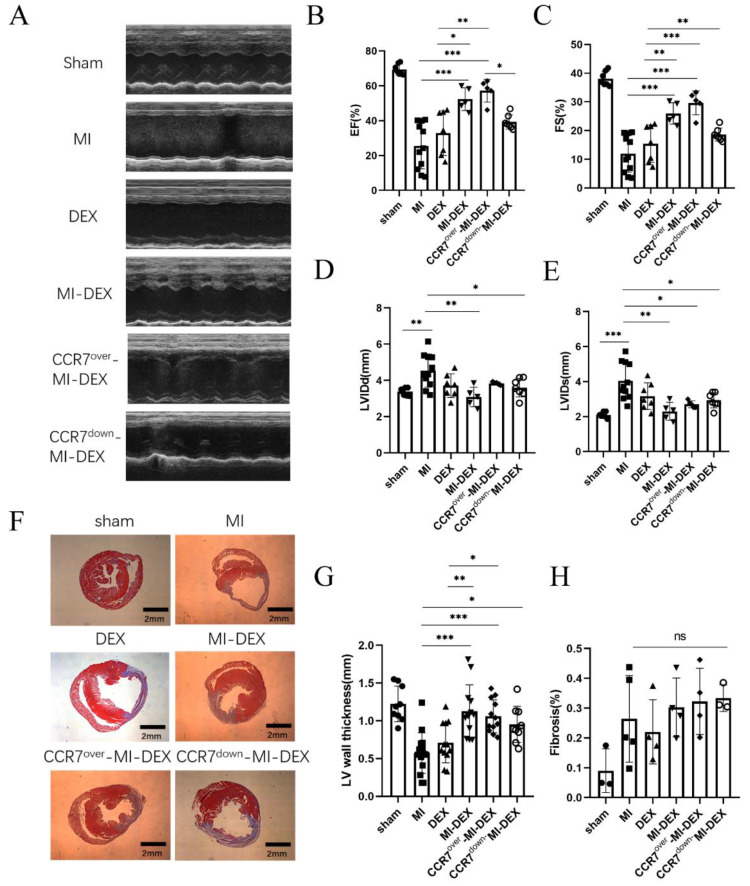
Enhancing MI-DEXs’ migration improved cardiac function after MI. Images of MI-model mice analyzed with echocardiography in different groups (**A**). Analysis of left ventricular ejection fraction (LVEF) in different groups (**B**). Analysis of left ventricular fraction shortening (LVFS) (**C**). Analysis of left ventricular end-diastolic diameter (LVIDd) (**D**). © Analysis of left ventricular end-systolic diameter (LVID©) (**E**). *n* = 5–10. Representative images analyzed by Masson’s trichrome staining in different groups (**F**). Infarct thickness (mm) in different groups (**G**). For each section, three measurements of different parts of the infarct area were taken for analysis. Analysis of fibrosis (%) in different groups (**H**). *n* = 3–5, * *p* < 0.05, ** *p* < 0.01, *** *p* < 0.001.

**Figure 4 pharmaceutics-15-00461-f004:**
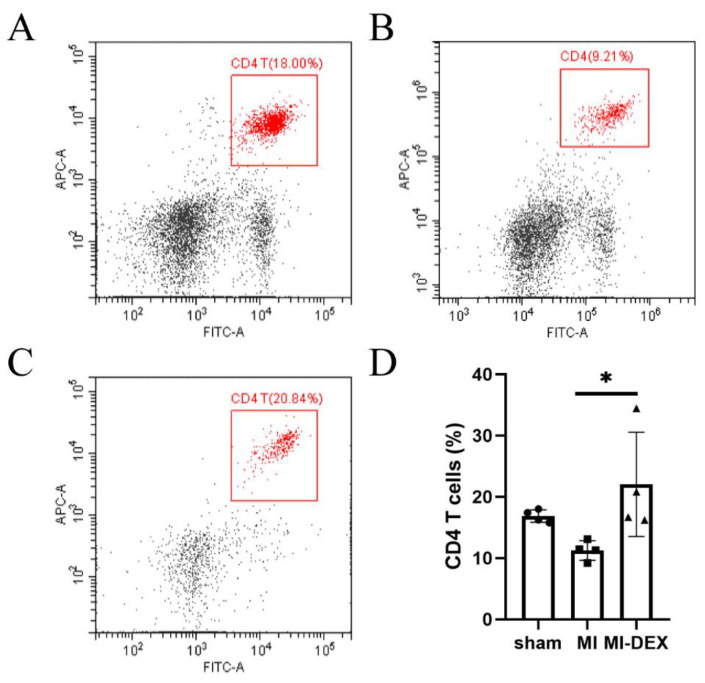
MI-DEXs raised the CD4^+^ T cell percentage in the spleen. Representative image of CD4^+^ T cells analyzed by flow cytometry in the sham group (**A**), MI group (**B**), and MI-DEX group (**C**); FITC-CD3; APC-CD4. Analysis of the proportions of CD45+, CD3+, and CD4+ T cells (**D**). *n* = 4, * *p* < 0.05.

**Figure 5 pharmaceutics-15-00461-f005:**
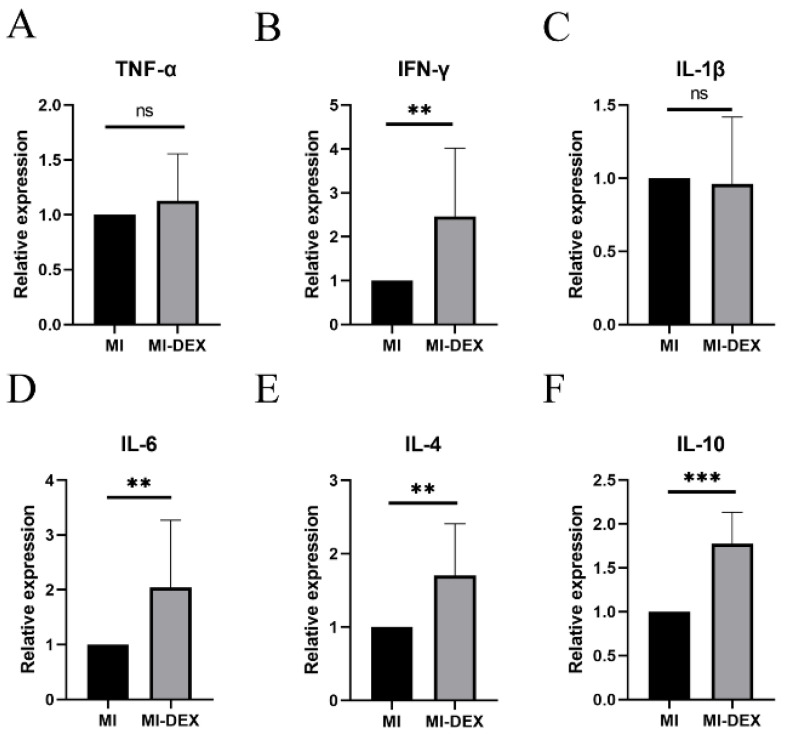
MI-DEXs increase anti-inflammatory cytokine expression in splenic CD4^+^ T cells. mRNA expression fold changes of different cytokines in different groups (**A**–**F**). *n* = 6–12, ** *p* < 0.01, *** *p* < 0.001; ns, no significance.

## Data Availability

The data sets used and/or analyzed during the current study can be obtained from the corresponding author if reasonably requested.

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
