# Peer review of "CCR7 Mediates Dendritic-Cell-Derived Exosome Migration and Improves Cardiac Function after Myocardial Infarction"

_pharmaceutics, 2023, doi:10.3390/pharmaceutics15020461_

Round 1
Reviewer 1 Report
This manuscript focusses on the role of dendritic cell derived exosomes. The authors demonstrate that dendritic cell derived exosomes improve myocardial function. Furthermore, they show that these dendritic cell derived exosomes migrate mediated by CCR7 and that they increase anti inflammatory cytokines in CD4+ T-cells.
The dendritic cell derived exosomes migrate by the interaction of CCR7 with its ligands CCL-19 and CCL-21 which are both highly expressed in the spleen.
This is overall an important study which moves the field forward containing very interesting novel data.
Major concern is the lack of detail in the method section. Especially in 2.3 and 2.5.
In 2.3 it is not clear how the dendritic cells are being harvested. How many dendritic cells are harvested? How are the newly harvested dendritic cells characterized?
In 2.5 (I) what type of primary cardiomyocytes are being used? What type of supernatant is used from them? Are the cardiac myocytes stimulated to change the supernatant?
(II)what supernatant from infarcted mice myocardium? How do you harvest the supernatant of the infarcted myocardium? Which cell types contribute to the supernatant if you use an entire mouse heart? Or do you only use the infarcted area? Please clarify
Minor concern are some grammatical errors: “Our previously study” should be our previous study (Lines 35, 210, 259, 277)
In line 50 remove “way of” from ”by way of CD4+ T cells
Author Response
Dear reviewer,
We would like to thank you for your careful reading, helpful comments, and constructive suggestions, which have significantly improved the presentation of our manuscript. We have carefully considered all your comments and revised our manuscript. The revised portions are marked in red, and our point-to-point responses to the comments are given below. Our revised version was edited by MDPI’s English editing service—the certification is attached in the attachment.
Response to Reviewer:
Overall: This is overall an important study which moves the field forward containing very interesting novel data.
Response: We thank you very much for carefully reading our paper and giving the above positive comments.
Q1: How many dendritic cells are harvested? How are the newly harvested dendritic cells characterized?
Response1: We gratefully appreciate the above suggestion. Bone marrow-derived (BM) cells were cultured at 106 cells/ml and BMDCs were determined using anti-mouse CD11c, CD40, CD80, and CD86 antibodies via flow cytometry. In the revised manuscript, we revised and added details in the Materials and Methods subsection 2.3, and the revised parts are marked in red.
Q2: what type of primary cardiomyocytes are being used? What type of supernatant is used from them? Are the cardiac myocytes stimulated to change the supernatant?
Response2: Thank you very much for pointing out these problems in our manuscript. We are very sorry that we did not provide the details in the method. We used three methods to mimic the MI microenvironment, which has been described in detail in our previous study (Ref.1): (I) the supernatants of hypoxic primary rat cardiomyocytes; (II) the supernatants of infarcted mouse myocardium; (III) the supernatants of necrotic HL‑1 cells. All three methods have the similar effect of activating DCs in mice to a similar extent compared to MI. In the present study, we used the supernatants of necrotic HL‑1 cells to mimic the MI microenvironment. Briefly, we washed HL-1 cells three times with serum-free medium to remove cell-derived exosomes and resuspended in serum-free medium with 107 cells/mL, then processed with five cycles of freezing in liquid nitrogen followed by thawing at 37 °C. We removed cell-membrane particles by centrifugation at 1,500 x g at room temperature for 30 min. Finally, we collected the supernatants of the necrotic HL‑1 cells for the following experiments. In the edited manuscript, we revised and added some details in the Materials and Methods subsection 2.5, and the revised parts are marked in red.
Q3: what supernatant from infarcted mice myocardium? How do you harvest the supernatant of the infarcted myocardium? Which cell types contribute to the supernatant if you use an entire mouse heart? Or do you only use the infarcted area? Please clarify.
Response3: Thank you very much for your comments. Supernatants from infarcted mouse myocardium were obtained from MI-model mice after ligation of the LAD artery for 8 h, which was described in detail in the previous study (Ref.1). In brief, after successful made the mouse MI-model for 8h, we harvested the hearts of the mice under sterile conditions, and we excised sections of the infarcted area and washed three times in PBS to remove blood cells. Following this, we cut the sections into small pieces, which were used to generate single-cell suspensions via the Gentle MACS system. Finally, we obtained the supernatants by centrifugation (1500g x 30min). According to experiment protocol, the infarcted cardiomyocytes mainly contributed to the supernatants.
Q4: Minor concern are some grammatical errors: “Our previously study” should be our previous study (Lines 35, 210, 259, 277), in line 50 remove “way of” from” by way of CD4+ T cells
Response4: Thank you very much for pointing out these grammar errors. The revised version has been edited by MDPI’s English editing service.
References:
- H. Liu, W. Gao, J. Yuan, C. Wu, K. Yao, L. Zhang, L. Ma, J. Zhu, Y. Zou, J. Ge, Exosomes derived from dendritic cells improve cardiac function via activation of CD4(+) T lymphocytes after myocardial infarction, Journal of molecular and cellular cardiology, 91 (2016) 123-133.
- Y. Maekawa, N. Mizue, A. Chan, Y. Shi, Y. Liu, S. Dawood et al., Survival and cardiacremodeling after myocardial infarction are critically dependent on the host innate immune interleukin-1 receptor-associated kinase-4 signaling: a regulator of bone marrow-derived dendritic cells, Circulation 120 (2009) 1401–1414.

Reviewer 2 Report
This is a basic study, which aimed to examine the role of CCR7 and its ligands CCL19/CCL21 in DEXs migration and the crosstalk between DEXs migration and its therapeutic effects on cardiac function after MI, to provide a new candidate for protecting cardiac function after MI. The authors concluded that DCs derived exosomes migration to spleen after MI was mediated by CCR7, and the mechanism of MI-DEXs improvement of cardiac function was partly activated of CD4+ T cells and secretion of IL-4 and IL-10, indicating the innovative way that improvement of cardiac function by enhancing the migration ability of MI-DEXs after MI, as a potential candidate of CCR7 for MI-DEX bioengineering to enhance its migration and improvement of cardiac function after MI. This reviewer considers that this paper might be interesting, but has several major concerns as described below.
Major comments:
1. First of all, this paper had too many typos. They should be corrected.
2. What was “BMDCs”? The abbreviation was lacked.
3. 3.1 Results section. The authors started Figure 1 D-E, but they should mention about Figure 1A-C first.
4. Figure 1D-E showed serum data, and Figure 1H-J showed spleen data. In spleen data (Figure 1I-J), the MI-7d data was lacked. Why?
5. Figure 4A-C. Which were which? They were not clear which data. Were they sham, MI, and MI-DEX? Through the whole manuscript, Figure description was not good.
6. Figure 5. Which organ did the authors use in this experiment?
Author Response
Dear reviewer,
We are grateful for your valuable suggestions and comments. We have carefully revised our manuscript according to your comments and suggestions. The revised portions are marked in red, and our point-to-point responses to your comments are given below. Our revised version was edited by MDPI’s English editing service—the certification is attached in the attachment.
Q1: First of all, this paper had too many typos. They should be corrected.
Response: Thank you very much for pointing out this problem in the manuscript. Our revised version was edited by MDPI’s English editing service. The certification is located in the attached files.
Q2: What was “BMDCs”? The abbreviation was lacked.
Response: Thank you for pointing out this problem. BMDCs is the abbreviation of bone marrow-derived dendritic cells. We have added the abbreviation in subsection 2.1 of the Materials and Methods section (line 68, marked in red).
Q3: 3.1 Results section. The authors started Figure 1 D-E, but they should mention about Figure 1A-C first.
Response: Thank you very much for your advice. We added the description about Figure 1A-C in subsection 3.1 in the Results section (from line 187, marked in red).
Q4: Figure 1D-E showed serum data, and Figure 1H-J showed spleen data. In spleen data (Figure 1I-J), the MI-7d data was lacked. Why?
Response: Thank you very much for your comments. Because we found there was no difference between the expressions of CCL19 and 21 in the serum MI-7d between different groups, we did not examine the expression of CCL19 and 21 in the spleen at 7d post-MI in the following experiments.
Q5: Figure 4A-C. Which were which? They were not clear which data. Were they sham, MI, and MI-DEX? Through the whole manuscript, Figure description was not good.
Response: Thank you very much for your comments. They were sham, MI, and MI-DEX groups. We revised the Figure 4 description and marked it in red. Furthermore, we checked all the figure descriptions, and the language was checked by MDPI’s English editing service.
Q6: Figure 5. Which organ did the authors use in this experiment?
Response: Thank you very much for your comments. CD4+ T cells were extracted from the mouse spleen of different groups; the detailed method description was updated in Materials and Methods subsection 2.3 and marked in red.

Round 2
Reviewer 2 Report
This reviewer has no further comment.